# A Case of Craniocervical Junction Arteriovenous Fistulas with a Brainstem Mass Lesion on Imaging: Case Report and Literature Review

**DOI:** 10.3390/brainsci13050839

**Published:** 2023-05-22

**Authors:** Zheng Peng, Yunfeng Wang, Cong Pang, Xiaojian Li, Zong Zhuang, Wei Li, Chunhua Hang

**Affiliations:** 1Department of Neurosurgery, Nanjing Drum Tower Hospital, Affiliated Hospital of Medical School, Nanjing University, Nanjing 210029, Chinawei.li@nju.edu.cn (W.L.); 2Neurosurgical Institute, Nanjing University, Nanjing 210029, China; 3Department of Neurosurgery, The Affiliated Huai’an No.1 People’s Hospital of Nanjing Medical University, Huai’an 223300, China

**Keywords:** intracranial mass lesions, craniocervical junction arteriovenous fistulas, digital subtraction angiography

## Abstract

Intracranial mass lesions occur within the cranial cavity, and their etiology is diverse. Although tumors and hemorrhagic diseases are the common causes, some rarer etiologies, such as vascular malformations, might also present with intracranial mass lesion manifestations. Such lesions are easily misdiagnosed due to the lack of manifestations of the primary disease. The treatment involves a detailed examination and differential diagnosis of the etiology and clinical manifestations. On 26 October 2022, a patient with craniocervical junction arteriovenous fistulas (CCJAVFs) was admitted to Nanjing Drum Tower Hospital. Imaging examinations showed a brainstem mass lesion, and the patient was initially diagnosed with a brainstem tumor. After a thorough preoperative discussion and a digital subtraction angiography (DSA) examination, the patient was diagnosed with CCJAVF. The patient was cured using interventional treatment, and an invasive craniotomy was not required. During diagnosis and treatment, the cause of the disease might not be apparent. Thus, a comprehensive preoperative examination is very important, and physicians need to conduct the diagnosis and differential diagnosis of the etiology based on the examination to administer precise treatment and reduce unnecessary operations.

## 1. Introduction

Since the cranium is a relatively closed space, intracranial mass lesions often displace normal tissues, damage focal nerves, and increase intracranial pressure [1,2]. These lesions are often caused by tumors but may also appear in hemorrhagic diseases, abscesses, cysts, parasitic infections, and vascular malformations [3]. To eliminate the cause of the disease, in addition to symptomatic treatment, other treatment methods, such as drainage, removal of the occupancy, and anti-infection, need to be administered. When intracranial mass lesions are present, the diagnosis and differential diagnosis of the etiology are very important, and the clinical management is variable for different etiologies [4,5]. A craniotomy is an invasive treatment modality, which cannot be applied to all intracranial mass lesions. Some rarer etiologies, such as vascular malformations, might also present with intracranial mass lesions and require the identification of clinical manifestations to avoid misdiagnosis. A patient with craniocervical junction arteriovenous fistulas (CCJAVFs) located on the brainstem was admitted to Drum Tower Hospital, affiliated with Nanjing University Medical School, on 26 October 2022. In this study, we summarized the diagnosis and treatment of this case and reviewed previously published studies. Our findings might help to better understand the etiology and diagnoses of intracranial mass lesions in the treatment of CCJAVF.

## 2. Case Presentation

The patient (34-year-old male) was admitted to the hospital with “intermittent headache with hoarseness for 10 days”. The patient experienced intermittent headache 10 days prior, which was intense during the attack but subsided after taking rest, accompanied by hoarseness. The patient did not experience dizziness, nausea, vomiting, weakness of the limbs, twitching of the limbs, difficulty in swallowing, or choking while drinking. A physical examination showed that his consciousness was clear, and his speech was fluent but slightly hoarse. Both pupils (2.5 mm in diameter) were equal in size and round and responsive to light, the muscle strength of the limbs was grade V, and the muscle tone was normal. The patient was admitted to the hospital and underwent routine imaging tests. Computed tomography (CT) examinations suggested an abnormal brainstem density (Figure 1A), and CT angiography (CTA) examinations showed no significant abnormalities (Figure 1B). We tentatively diagnosed the cause as a brainstem tumor and initially suggested craniotomy to remove the mass lesion. The cranial magnetic resonance imaging (MRI) scan showed lamellar, slightly long T1 and slightly long T2 signal intensities in the brainstem, with a maximum area of about 19 mm × 19 mm in the transverse position. The area of the lesion increased significantly after enhancement (Figure 1C–G). The Flair sequence showed a strong signal (Figure 1H). The MRI examinations suggested the presence of a lamellar abnormal signal in the brainstem with a vascular flow space shadow. There were prominent vascular markings anterior and posterior to the brain stem on MRI (high signal with contrast and voids on T2 images), which pointed to the possibility of AVF.

We ruled out the possibility of an intracranial hemorrhagic lesion, which is often associated with increased intracranial pressure, severe headache, or loss of consciousness [6,7,8]. To further clarify the diagnosis, we performed a digital subtraction angiography (DSA) examination, which showed that our preoperative diagnosis was inaccurate and the mass lesion was not caused by a tumor. DSA examinations revealed CCJAVFs, where the right ascending pharyngeal artery acted as the donor artery, and the spinal veins and the occipital cortical veins acted as the drainage veins (Figure 2A–C). The treatment modality should be decided by the anatomical structure and the position of AVF, for which the operation or the intravascular intervention was suitable. The DSA examination suggested that a craniotomy would be unnecessary and that this patient only required management of the arteriovenous fistula. Based on these preoperative findings, an endovascular intervention was performed. A 6F guide sheath was placed through a puncture in the right femoral artery, and a 6F guiding catheter was applied for imaging. The 6F guide catheter was placed at the beginning of the right external carotid artery with the assistance of a guidewire under the roadmap. The contrast media determined that there was no spasm of the vessel. A microcatheter was placed into the site close to the fistula under the road map for embolization, and a spring coil was placed in the superior trunk of the right ascending pharyngeal artery to reduce blood flow and further thrombotic occlusion (Figure 3A,B). The angiogram showed a significant decrease in the blood flow in the superior trunk of the ascending pharyngeal artery. Then, an Onyx biologic gel was injected slowly and intermittently into the upper and lower trunks of the ascending pharyngeal artery through the microcatheter (Figure 3C,D). The diffusion and regurgitation of the Onyx biologic gel were repeatedly observed via microcatheter imaging until the branch at the fistula was completely embolized. The angiogram showed a complete non-visualization of the supply artery and drainage vein, and all intracranial branches were present. Five days after surgery, the DSA analysis showed a complete embolization of the fistula (Figure 4A). The CT examinations showed a lamellar, slightly hypointense shadow in the brainstem, without hemorrhage or ischemia (Figure 4B). The MRI examinations showed lamellar, slightly long T1 and slightly long T2 signals in the brainstem, with a maximum cross-section of about 10 mm × 8 mm. The brainstem mass lesion after treatment was reduced when compared to that before treatment (Figure 4C,D). It is important to note that embolization may pose an additional risk. Dimethyl sulfoxide injection is required prior to embolization and may have potential neurotoxicity. Access vascular injury may occur during embolization. If the biogel enters a vessel other than the lesion site, it may cause cerebral infarction. During embolization, hemodynamic changes also require attention. In fact, immediately after the patient’s surgery was completed, we transferred the patient to the ICU for detailed monitoring as well as an assessment of the patient’s status. After 7 days of monitoring, we confirmed that the patient had no complications and then transferred him to the general ward. The patient recovered well after the operation, and his headaches and hoarseness decreased. He was discharged with a Karnofsky score of 90.

## 3. Discussion

Tumors are the most common cause of intracranial mass lesions, such as gliomas, auditory neuromas, and intracranial metastases [9,10]. When an intracranial tumor grows to a certain volume, the intracranial occupying effect or neurological impairment worsens and becomes a typical symptom of tumorigenesis. Cerebral hemorrhage is also a common cause of intracranial mass lesion. It has an acute onset and is accompanied by severe neurological impairment [11]. Physicians should not only focus on the common causes of intracranial mass lesion but should consider all factors that might contribute to the disease. Considering that the intracranial mass lesion in this patient was caused by CCJAVFs, we thought that a vascular disease might be the cause of the intracranial mass lesion. Several studies have reported intracranial mass lesions caused by vascular abnormalities. For example, Khan et al. reported the case of a 45-year-old male who presented with rapidly progressive severe attention and memory deficits over a week, with an initial diagnosis of thalamic edema and venous embolism, which was eventually confirmed to be a brain dural arteriovenous fistula (DAVF). The nerve functions of the patient were restored after treatment [12]. Hanyu et al. reported the case of a 45-year-old male with a headache and a final angiographic diagnosis of DAVF, although he was initially suspected of having brain malignancy based on the results of positron emission tomography-CT (PET-CT) and MRI [13]. These cases have some similarities. For example, such patients have an acute onset and rapid progression, and clinical symptoms are mostly caused by occupying effects. However, these manifestations are nonspecific and are often ignored by clinicians. Intracranial mass lesions caused by arteriovenous malformations are often initially misdiagnosed, which might require angiography before treatment for a definitive diagnosis of arteriovenous malformations. Intracranial mass lesions caused by arteriovenous fistula might be associated with an increase in the size of the lesion due to primary changes, and clinical symptoms might be effectively relieved via timely treatment. In contrast, intracranial mass lesions caused by some vascular diseases might be caused by secondary injury to the nerve tissue, causing a tumor-like lesion effect. As it is a secondary change, its diagnosis is more challenging, and, thus, this condition often leads to a relatively poor prognosis.

CCJAVFs are relatively rare in clinical practice and are caused by the formation of fistulas in the arteriovenous region of the craniocervical junction, which can occur when inflammation, trauma, or other factors lead to disturbances in the arteriovenous pressure gradient and the obstruction of venous return. The etiology is not clear and might be related to congenital anatomical abnormalities, trauma, thrombosis, or medically induced injuries [14,15]. Due to the specific site of occurrence, the lesion is often located close to the craniocervical junction, and the clinical manifestations of CCJAVFs are diverse. Clinical examinations based on the effects of cranial mass lesions can easily lead to misdiagnosis because of the lack of clinical manifestations of the primary disease. The clinical symptoms of our patient were mainly headache and hoarseness. Headache is not an idiosyncratic symptom, and many craniocerebral diseases can cause headaches. Hoarseness might be caused by abnormalities in the laryngeal recurrent nerve [16], which is a branch of the vagus nerve. Routine CT and MRI examinations can be performed to detect occupancy at the site of the lesion. The clinical symptoms and imaging results of occupancy effects may not show any signs of a cerebral hemorrhage, such as loss of consciousness, hemiparesis, and severe headaches in patients [7,8,17], thus increasing the chances of misdiagnosing the disease as a brain tumor. Our patient had an acute onset and a short duration of illness, which indicated a clinical manifestation that was different from a tumor. Additionally, CCJAVFs can also affect the spinal cord, causing problems, such as numbness in the extremities, difficulty urinating, and hemiparesis [18,19]. Arteriovenous malformations need to be considered when diagnosing the etiology of craniocerebral mass lesions. During diagnosis and treatment, the clinical manifestations of patients are complex, and their disease etiology often goes unnoticed. Some etiologies might only be found when reviewing patients who have failed treatment. When some clinical manifestations occur that do not match the description of a cranial tumor or hemorrhage, even if the presentation is not typical, a detailed examination is needed to exclude other causes of the disease.

In the clinical setting, common etiologies of intracranial mass lesions and the underlying etiology need to be investigated. Only by improving the efficacy of preoperative examinations can the rate of misdiagnosis be reduced. CCJAVF has a complex vascular configuration, lacks typical symptoms, and is easily misdiagnosed on imaging. CTA was performed during admission and did not suggest vascular abnormalities. Although CTA is less accurate than DSA for vascular diseases, it is widely used as a preliminary screening tool for vascular diseases. As CTA cannot detect vascular abnormalities, vascular diseases cannot be ruled out following a CTA examination. Additionally, CTA does not have a high detection rate for arteriovenous fistulas (AVFs). The diagnosis of AVF is not complicated, and DSA is the gold standard for diagnosis [20]. Studies have also performed adjunctive diagnosis using indocyanine green [21,22,23] or performing MRI [24], but for most clinical centers, DSA is the most reliable and convenient diagnostic modality. DSA can detect the supply artery and the draining vein of the arteriovenous fistula. In our patient, we found that the right ascending pharyngeal artery served as the supply artery, and the spinal vein and the occipital cortical vein served as the draining vein.

To effectively treat AVF, the venous drainage of the fistula needs to be occluded, and a normal venous flow needs to be restored. The main treatment modalities for AVF include microsurgery, which allows for direct treatment of the fistula [25], and interventional surgery, which is less invasive and has a higher recanalization rate than microsurgery [26]. However, with the continuous development of embolization materials, the rate of complete embolization with interventional treatment has increased [27,28]. The disease lasted for a short duration, and a thorough preoperative examination also provided the basis for disease management. We adopted endovascular interventions to treat the arteriovenous fistula, and the DSA examinations showed that the abnormal blood flow was occluded. The patient recovered well after the procedure; his headache decreased, and the hoarseness improved. The early management of CCJAVF is important, and the prognosis of a patient is often related to the degree of neurological involvement.

Multi-disciplinary treatment (MDT) is very important. In modern medicine, medical disciplines are more finely classified to provide targeted treatment for diseases, and specialists might only be familiar with a small field. Our patient was initially treated by an oncology specialist who was not aware of the vascular abnormality and thought it might be a manifestation of a tumor. During a comprehensive neurosurgery consultation, the vascular specialist identified the case and recommended a DSA examination to clarify the relationship between the vessel and the mass lesion, which was eventually diagnosed as an AVF. This suggested that during diagnosis and treatment, if certain symptoms do not match the field of expertise of a specialist, multi-disciplinary discussions involving other specialists need to be conducted at the earliest possible moment.

## 4. Conclusions

To summarize, a cranial mass lesion is a very common clinical syndrome that mostly starts with impaired neurological functions, and identifying the etiology of a cranial mass lesion is important. We should pay attention to the clues that would lead to considering AVF. Although most intracranial mass lesions are caused by tumors or hemorrhage, we reported the case of a brainstem mass lesion caused by CCJAVF. Our findings suggested that intracranial mass lesions might also be caused by arteriovenous malformations. Cranial mass lesions caused by arteriovenous malformations might be misdiagnosed as tumors, especially when the patient mostly has neurological involvement and lacks the typical symptoms of vascular malformations. Thus, to avoid misdiagnosis, clinicians need to differentiate arteriovenous malformations or other types of vascular diseases when the clinical manifestations of intracranial mass lesions are unclear or when certain clinical manifestations do not match the typical symptoms of common causes. DSA examinations need to be performed during clinical treatment to reduce the rate of misdiagnosis.

## Figures and Tables

**Figure 1 brainsci-13-00839-f001:**
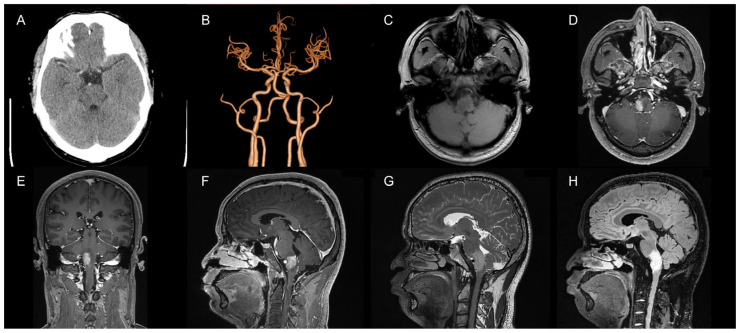
Preoperative imaging. (**A**) A CT examination suggested abnormal brainstem density; (**B**) CTA showed no significant abnormalities; (**C**) Abnormal brainstem signal in the T1 sequence; (**D**) T1 enhancement suggested significant enhancement at the site of the lesion (axial view); (**E**) T1 enhancement suggested significant enhancement of the focal area (coronal view); (**F**) T1 enhancement suggested significant enhancement at the site of the lesion (sagittal view); (**G**) A T2 sequence with a slightly long signal in the brainstem and vascular flow space shadow in the focal area; (**H**) The Flair sequence suggested a high signal at the focal area.

**Figure 2 brainsci-13-00839-f002:**
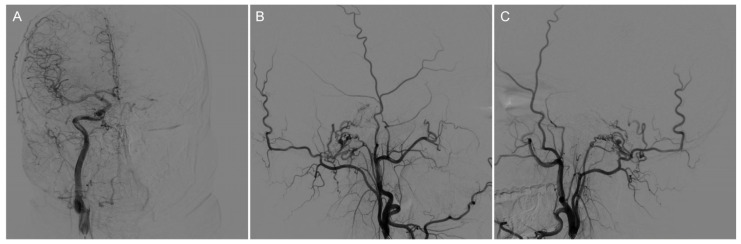
Preoperative DSA of the AVF showed that the right ascending pharyngeal artery served as the donor artery, and the spinal veins and occipital cortex veins served as the draining veins. (**A**) Preoperative imaging (frontal view); (**B**) Preoperative imaging (right-sided view); (**C**) Preoperative imaging (left-sided view).

**Figure 3 brainsci-13-00839-f003:**
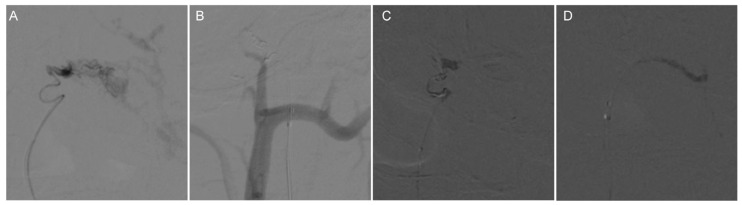
Intraoperative DSA angiographic treatment of CCJAVF. (**A**) Microcatheterization through the superior trunk of the ascending pharyngeal artery to reach the lesion; microcatheterization was performed to clarify the fistula. (**B**) Spring coil embolization of the fistula via the microcatheter; (**C**) The first microcatheter located in the superior trunk of the ascending pharyngeal artery was used to occlude the fistula with a biologic gel; (**D**) The second microcatheter in the lower trunk of the ascending pharyngeal artery was used to occlude the fistula with a biologic gel.

**Figure 4 brainsci-13-00839-f004:**
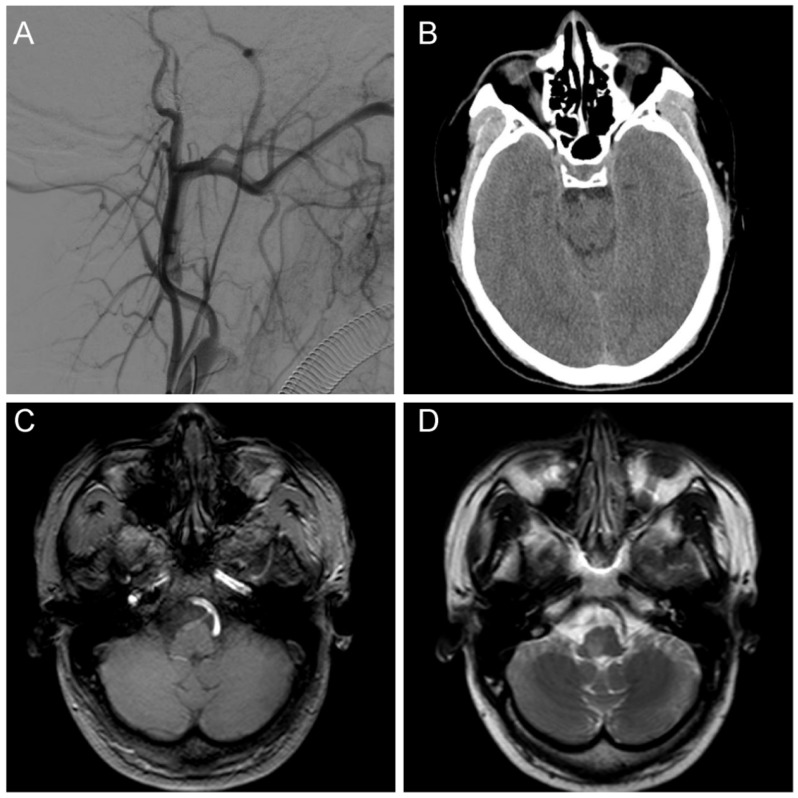
Postoperative examination. (**A**) A postoperative DSA examination indicated complete fistula occlusion; (**B**) A CT examination indicated no hemorrhage and ischemia at the site of the lesion; (**C**) The T1 sequence of MRI showed a slightly long signal area in the brainstem without abnormal postoperative injuries; (**D**) The T2 sequence of MRI showed a slightly long signal area in the brainstem with a normal cranial structure.

## Data Availability

The data in this study can be obtained from the article.

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
