# Peer review of "A Case of Craniocervical Junction Arteriovenous Fistulas with a Brainstem Mass Lesion on Imaging: Case Report and Literature Review"

_brainsci, 2023, doi:10.3390/brainsci13050839_

Round 1

Reviewer 1 Report

Some of the English is non idiomatic. Antidiastole is a term rarely used in English language papers: perhaps differential diagnosis would be better. 

The repeated use of the term "cranial occupancy" is also intrusive to the reader. This term is not commonly used by native English speakers; it would be more common to refer to a mass lesion, but at least the term is self explanatory and is not confusing.

The case is important as AVFs may be misdiagnosed and this case exemplifies some of the difficulties. It would be important to highlight the clues that should lead to considering AVF. In this case there are prominent vascular markings anterior and posterior to the brain stem on MRI (high signal with contrast and voids on T2 images) which point to the possibility of AVF. This could be emphasised more.

There is some unnecessary discussion about how to manage brain tumour which is irrelevant to this case. The sentences "The clinical management of the different etiologies causing intracranial occupancies 202 is different. For patients with tumors, a craniotomy needs to be performed. If a patient is 203 diagnosed with a brain tumor, craniotomy is required to obtain the tissue of the lesion. 204 The tissue is subsequently tested pathologically and staged. A craniotomy is traumatic for 205 the patient, and those who undergo a craniotomy require rigorous postoperative care to 206 facilitate incisional recovery. If a patient has diabetes, hypoproteinemia, or cachexia, the 207 craniotomy wound heals slowly, and some patients might even show signs of delayed 208 healing or no healing. We thoroughly examined the patient and found that his intracranial 209 occupancy was caused by an AVF, and thus, we did not perform a craniotomy." could easily be omitted.

Likewise, in the first paragraph of discussion there is reference to a range of conditions such as malignant MCA infarcts and cerebral angiitis which have little relevance to this case.

Author Response

Thank you very much for your comments. Please note that any revisions to the manuscript have been marked up using the “Track Changes” function.

1. Some of the English is non idiomatic. Antidiastole is a term rarely used in English language papers: perhaps differential diagnosis would be better.

        Response: Thank you very much for your suggestion. We replaced "antidiastole" with "differential diagnosis" in the manuscript.

        Modification:

We replaced "antidiastole" with "differential diagnosis" in the manuscript: line 17, line 25, line 39.

2. The repeated use of the term "cranial occupancy" is also intrusive to the reader. This term is not commonly used by native English speakers; it would be more common to refer to a mass lesion, but at least the term is self explanatory and is not confusing.

Response: Thank you very much for your comment. We used "mass lesion" instead of "occupancy" to make the manuscript more understandable.

Modification:

We used "mass lesion" instead of "occupancy" in the manuscript: line 3, line 13, line 15, line 20, line 28, line 32, line 38, line 41, line 42-43, line 48-49, line 77, line 104, line 142, line 146, line 148, line 149-150, line 151, line 152, line 162-163, line 165, line 168, line 169, line 184, line 197-198, line 204, line 246, line 251, line 255, line 256, line 257, line 262.

3. The case is important as AVFs may be misdiagnosed and this case exemplifies some of the difficulties. It would be important to highlight the clues that should lead to considering AVF. In this case there are prominent vascular markings anterior and posterior to the brain stem on MRI (high signal with contrast and voids on T2 images) which point to the possibility of AVF. This could be emphasised more.

Response: We strongly agree with you and have emphasised this thesis in the article.

Modification:

“We tentatively diagnosed the cause as a brainstem tumor and initially suggested craniotomy to remove the mass lesion” (line 62-63)

 “There were prominent vascular markings anterior and posterior to the brain stem on MRI (high signal with contrast and voids on T2 images) which pointed to the possibility of AVF” (line 69-71)

“To further clarify the diagnosis, we performed the digital subtraction angiography (DSA) examination, which showed that our preoperative diagnosis was inaccurate and the mass lesion was not caused by a tumor.”(line 75-78)

“We should pay attention to the clues that would lead to considering AVF.” (line 253-254)

4. There is some unnecessary discussion about how to manage brain tumour which is irrelevant to this case. The sentences "The clinical management of the different etiologies causing intracranial occupancies 202 is different. For patients with tumors, a craniotomy needs to be performed. If a patient is 203 diagnosed with a brain tumor, craniotomy is required to obtain the tissue of the lesion. 204 The tissue is subsequently tested pathologically and staged. A craniotomy is traumatic for 205 the patient, and those who undergo a craniotomy require rigorous postoperative care to 206 facilitate incisional recovery. If a patient has diabetes, hypoproteinemia, or cachexia, the 207 craniotomy wound heals slowly, and some patients might even show signs of delayed 208 healing or no healing. We thoroughly examined the patient and found that his intracranial 209 occupancy was caused by an AVF, and thus, we did not perform a craniotomy." could easily be omitted.

       Response: Thank you very much for your suggestion, we have removed this section in the revised version.

5. Likewise, in the first paragraph of discussion there is reference to a range of conditions such as malignant MCA infarcts and cerebral angiitis which have little relevance to this case.

Response: We strongly agree with your comment and have removed this section in the revised version.

Reviewer 2 Report

I think that this case report is far from acceptance.

The authors demonstrated clinical feature, radiological findings and management in craniocervical junction arteriovenous fistulas (CCJAVF). Some reports of small series about CCAVF had already been released. In those reports, clinical feature, radiological; findings and surgical management had been described in detail. So that this case report dose not have novelty. The authors should cooperate the novelty, clinical feature, radiological findings.

I am not able to agree with:

Although the authors described that they reduce unnecessary operation and intravascular intervention by embolization, but it should be decided by the anatomical structure and the position of AVF which the operation or the intravascular intervention was suitable. In depended on the case, direct surgery and processing of fistula using intraoperative angiography or other means.

I suggest that:

The authors should describe the material and procedure of embolization in detail.

Author Response

Thank you very much for your comments. Please note that any revisions to the manuscript have been marked up using the “Track Changes” function.

1.The authors demonstrated clinical feature, radiological findings and management in craniocervical junction arteriovenous fistulas (CCJAVF). Some reports of small series about CCAVF had already been released. In those reports, clinical feature, radiological; findings and surgical management had been described in detail. So that this case report dose not have novelty. The authors should cooperate the novelty, clinical feature, radiological findings.

        Response: Thank you very much for your comments. We agree with you that some reports of small series about CCAVF had already been reported. However, vascular malformations with cranial mass lesions as the initial presentation are uncommon and can easily be diagnosed as tumors. The preoperative examination and treatment of tumors and vascular malformations are completely different in clinical work. We describe this case in the hope of drawing the attention of clinicians, providing experience in clinical work, avoiding misdiagnosis and mistreatment, and thus saving medical resources.

2.Although the authors described that they reduce unnecessary operation and intravascular intervention by embolization, but it should be decided by the anatomical structure and the position of AVF which the operation or the intravascular intervention was suitable. In depended on the case, direct surgery and processing of fistula using intraoperative angiography or other means.

Response: Thank you very much for your comments. We agree with you and have revised the manuscript.

Modification:

“Treatment modality should be decided by the anatomical structure and the position of AVF which the operation or the intravascular intervention was suitable.” (line 80-82)

3.The authors should describe the material and procedure of embolization in detail.

        Response: Thank you very much for your comments. We have added the material and procedure of embolization in the manuscript.

        Modification:

        “A 6F guide sheath was placed by puncture in the right femoral artery and a 6F guiding catheter was applied for imaging. The 6F guide catheter is placed at the beginning of the right external carotid artery with the assistance of guidewire under the roadmap. The contrast media determined that there was no spasm of the vessel. A microcatheter was placed into the site close to the fistula under the road map for embolization. and a spring coil was placed in the superior trunk of the right ascending pharyngeal artery to reduce blood flow and further thrombotic occlusion (Fig. 3A & B). The angiogram showed a significant decrease in blood flow in the superior trunk of the ascending pharyngeal artery. Then, an Onyx biologic gel was injected slowly and intermittently into the upper and lower trunks of the ascending pharyngeal artery through the microcatheter (Fig. 3C & D). The diffusion and regurgitation of the Onyx biologic gel were repeatedly observed by microcatheter imaging until the branch at the fistula was completely embolized.” (line 85-97)

Reviewer 3 Report

The authors propose an interesting case of a brainstem arteriovenous fistula. The case, although nicely treated, is not unique in literature, with more than 30 original articles and reviews on this specific topic available in literature only in the last 3 years. A proper review has been already published this year, and overall, I can’t see any novelty in the paper or significant messages.    

Author Response

Thank you very much for your comments. Please note that any revisions to the manuscript have been marked up using the “Track Changes” function.

The authors propose an interesting case of a brainstem arteriovenous fistula. The case, although nicely treated, is not unique in literature, with more than 30 original articles and reviews on this specific topic available in literature only in the last 3 years. A proper review has been already published this year, and overall, I can’t see any novelty in the paper or significant messages.

        ResponseThank you very much for your comments. Some reports of small series about CCAVF had already been reported before. Nevertheless, we would very much appreciate a detailed description of this case because vascular malformations with cranial mass lesions as the initial presentation are uncommon and can easily be diagnosed as tumors. The preoperative examination and treatment of tumors and vascular malformations are completely different in clinical work. We describe this case in the hope of drawing the attention of clinicians, providing experience in clinical work, avoiding misdiagnosis and mistreatment, and thus saving medical resources.

Round 2

Reviewer 2 Report

This article improved by revision by the authors. The novelty of this case report became to understand easily. The authors properly answered and revised according to comment of the reviewer. I recommend that this article should be accepted. 

Author Response

-